# Evaluating post 2024 election scenarios for the UK based on political party manifestos

Richard Stafford[1], Jenny Alexander[2], Stephen Axon[1], Zach Boakes[1], Elena Cantarello[1], Abigail R. Croker[3,4], Marin Cvitanović[1], Victoria Dominguez Almela[5], Tilak Ginige[1], Natalie Harris[1], Ellie-Anne Jones[1], Yiannis Kountouris[3], Darren Lilleker[2], Freya Russell[1], James R. Sokolnicki[1], Sarah J. Upson[1], Ellie Vincent[1]

**1** Department of Life and Environmental Sciences, Bournemouth University, Poole, United Kingdom, **2** Department of Humanities and Law, Bournemouth University, Poole, United Kingdom, **3** Centre for Environmental Policy, Imperial College London, London, United Kingdom, **4** Centre for BioComplexity, High Meadows Environmental Institute, Princeton University, Princeton, New Jersey, United States of America, **5** School of Geography and Environmental Science, University of Southampton, Southampton, United Kingdom

## Abstract

Manifestos provide a vision for political parties to enact if elected to government. While manifestos may not be enacted in full, they provide some of the best information for the public in deciding how to vote. However, manifestos tend to contain both enactable policies (e.g., tax cuts) and outcomes or visions of what these policies may achieve for society (e.g., higher disposal income). These outcomes may be deliberately misleading, inaccurate or only a partial picture of how policies will materialise, as it is unlikely a single policy will influence a single outcome. This work creates a complex system model of the economic, societal and environmental landscape of the UK and assesses how it would be affected if political parties enacted their 2024 general election manifesto policies in full. Our model creates a more complete picture of how the UK may look under different parties, rather than examining the manifestos alone (using data solely from manifestos almost 30% of our model's node values had no information, falling to just 5% after running the models). The model also has the capacity to provide a holistic reflection of the parties manifesto plans, illustrating the impacts each party's policies could have, should they be enacted. Prior to integrated analysis the most right-wing of the parties studied; Reform UK, aligned strongly with the Conservative party, however, post analysis Reform became a clear outlier. We also demonstrate unintended, misreported or indirect effects of policies. Most notably, parties who had the strongest tax cutting policies resulted in lower average incomes and higher levels of inequality in society, despite the rhetoric provided for these policies in the party manifestos. The results demonstrate the ability to integrate multiple types of information across political, economic, environmental, and social landscapes to help visualise implications of policy and politics more widely.

**Data availability statement:** All relevant data are within the manuscript and its Supporting information files.

**Funding:** The author(s) received no specific funding for this work.

**Competing interests:** The authors have declared that no competing interests exist.

## 1. Introduction

The UK political landscape is complex, with multiple interconnected components, essentially forming a complex system [1,2]. While there are clear political and societal outcomes (including health, cost of living, crime rates, the environment and immigration); alongside clear policy levers (such as taxation, spending and legislation) understanding the pathways from policy levers to outcomes is complex and involves interplay between economic, environmental and social systems, amongst others [3,4].

Political party manifestos provide a vision of the outcomes the party would try to achieve if it was elected to form the next government [5]. In practice, manifestos are not usually fulfilled by the government in full, although evidence suggests fulfilment of pledges is greater than commonly perceived (e.g., [6,7]). This incomplete fulfilment may arise due to lack of commitment to the pledges made [8]; due to coalition or strong political, legal or public opposition to pledges [9]; or through strong, unexpected external influences which rapidly change priorities, such as the Covid-19 pandemic [10]. Equally, smaller political parties, unlikely to form governments, can create unrealistic or unachievable manifesto pledges, knowing they will never have to deliver these in full [8].

Despite the shortcomings of manifestos, they present the best insight into policies that political parties would enact if in government [5]. However, manifestos contain and interweave narratives on policies that they can enact (e.g., raising or lowering taxes) and outcomes that they are unable to directly control (e.g., crime rates). Often a direct cause and effect approach is applied between policies and outcomes in manifestos. For example, the Reform UK manifesto (or contract) [11], states a policy of: "Tax Relief of 20% on all Private Healthcare and Insurance". It then goes on to claim an outcome: "This will improve care for all by relieving pressure on the NHS. Those who rely on the NHS will enjoy faster, better care." Such statements seem logical, but the quality of health care through the publicly funded National Health Service (NHS) will in fact depend on multiple factors, not just subsidy of private provision. Equally, other claims in manifestos imply a cause and effect relationship, even without stating them explicitly. For example, in the Conservative manifesto [12]: "We will cut employee National Insurance to 6% by April 2027 […] a total tax cut of £1,350 for the average worker on £35,000." This implies financial benefit, yet if this tax cut is combined with a loss of benefits, or higher cost of living, the net effect could be to reduce average disposable income. Some manifesto content is also based on speculation, such as Labour's increase in total income tax revenue through economic growth, which they can not directly control, rather than through changes to the tax rate for individuals, which they can control [13]. In many cases, policies may be designed to achieve particular party ideologies (i.e., low taxation), but the outcomes may be stated to appeal to a wider audience, even if the relationship between the two lacks empirical evidence.

Typically, the relationship between a single policy and a societal, economic or environmental outcome is complex. For example, increasing police numbers may be shown to cut crime, but crime is also affected by multiple other issues, such as

poverty [14,15]. As such, there may be multiple policies which need to interact to achieve the required outcome. Equally, policies can have unintended or second order effects. For example, while it is frequently assumed there should be a positive environmental outcome from policies to accelerate the uptake of electric cars, components of batteries can cause negative environmental consequences through pollution [16]. The interconnected economic, environmental, social and political landscape needs to be considered as a complex system in order to make accurate predictions on how policies will holistically interact and achieve outcomes. While such approaches have been proposed more than a decade ago [1,17] and strong advocacy exists for this approach, few studies have been conducted using these approaches (e.g., [18,19] for environmental outcomes, [20] for economic applications and the UK Government's Department of Work and Pensions Policy Simulation Model for policy decisions linked to poverty and income distribution), and to the best of our knowledge, no studies until now have attempted to integrate the full economic, environmental, social and political landscape into a complex system model.

Here we study the manifestos of the five major political parties in England (many of which also exist in the UK's devolved nations) using a complex system model of the political, societal, environmental and economic landscape of the UK based on Bayesian belief networks. Using the model we assess: (1) the effectiveness of policies in political party manifestos in achieving societal, environmental and economic outcomes, (2) unintended or second order consequences of policies, and (3) create a prediction of the overview of the holistic outcomes different manifestos would achieve, if enacted in full. The prediction is based on the policy levers in each party's manifesto being enacted in full and cannot account for changes made to these pledges, external events and additional policies being enacted. As such, our predictions do face limitations, yet provide a more complete picture than consideration of the manifesto alone.

## 2. Methods

Our model builds on previous work by many of the present study authors in creating a holistic environmental policy/ outcome model [18]. However, this model is more ambitious in tackling the breadth of the entire political landscape, including more economic factors and more social factors along with the environmental factors considered previously. The model is based on modified Bayesian belief networks (BBNs), built and parameterised following the procedures in Dominguez Almela et al., and full details of model equations and parameterisation processes can be found in this paper [21]. An overview of the modelling process used here can be found in Fig 1, which references the below sections to provide more detail.

BBNs can be thought of as a series of 'nodes' which here represent policies or outcomes of policies, and 'edges' which connect the nodes through direct cause and effect interactions. When initial or 'prior' values of nodes are changed (i.e., a node for income tax is changed to represent a cut in income tax as a political policy), this change propagates through the network (via edges), changing values of interacting nodes (resulting in 'posterior' values). Complex outcomes can arise from the network as multiple cause and effect relationships are likely to exist for any given node. For example, a 'policing' node has a direct causal effect on a 'crime rates' node but is unlikely to be the only node affecting crime (e.g., cost of living or inequality may all affect crime).

### 2.1. Overview of the model

The model represents a simplified version of the political, economic, social and environmental landscape of the UK. The temporal aspect of the model represents around 5 years (or a single parliamentary term in office under the Dissolution and Calling of Parliament Act 2022) and is set by considering this timeframe in edge interactions. As an example of how part of the model works, economic processes, which underlie and interact with all other processes, are modelled through various forms of taxation (inflows to the budget) and spending (outflow). If our modelled outflows exceed inflows then deficits arise through negative values in the 'Treasury Funds' node (although we do not present a detailed 'accounting model' here, with values representing increases or decreases from the current situation). In our model, deficits do not feed back into our model

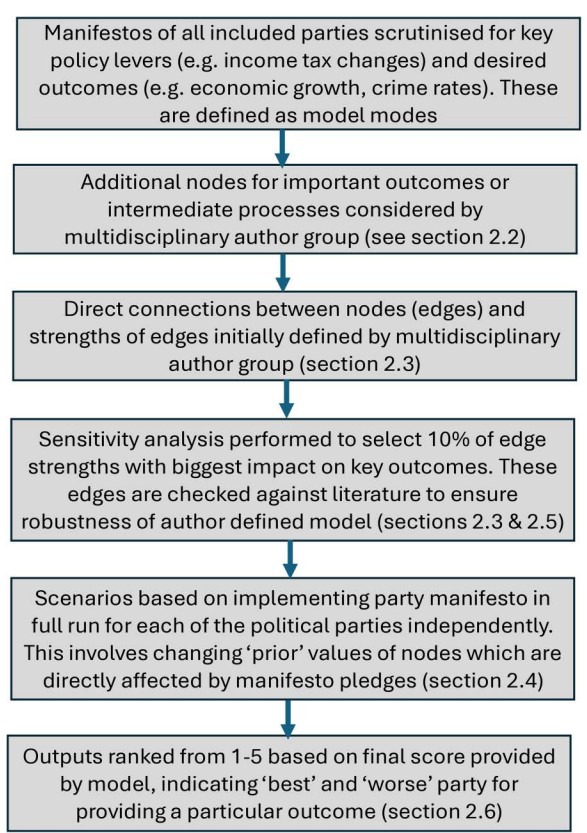

**Fig 1. Overview of the modelling process including node and edge selection and edge weighting.** The figure refers to specific sections of the text to provide more details.

to increase taxation or reduce spending, but financial markets, especially the bond market, can be negatively affected by deficits, lowering expectations that the government will be able to finance or repay deficits in the future [22]. This intentional lack of feedback may not represent reality but helps account for processes such as ideas from modern monetary theory (MMT), where deficits or increasing the production of money by central banks can be utilised, rather than a 'tax and spend' narrative [23]. Our models assume theories such as MMT may be valid, but equally we should be cautious of large negative values in our 'Treasury Funds' and 'Financial Markets' nodes, and we evaluate these as specific outputs of the model. Financial markets can also be affected by increases in some forms of taxation, nationalisation, and inflation, but respond positively to increased private investment, increases in income and employment [24,25]. Economic growth (here defined as a percentage of GDP) can be boosted by financial markets, investments in utilities, transport and private investment, and (directly) limited by some forms of taxation. However, the model represents a complex, interacting system. For example, while income tax rises may not directly affect economic growth, they may lower disposable income (although see results below for alternative scenarios), leading to reduction in economic growth as an indirect effect. Equally increasing budget deficits may have less effects on financial markets, if used for spending which leads to greater private investment [22].

The model also considers key social concerns such as housing, transport, water and sewage, employment and apprenticeships, policing, health and well-being (including the NHS), cost of living, legal immigration and asylum/ refugees, the environment (including climate and biodiversity), agriculture and pensions. Again, all of these are part of a complex system interacting with the economic aspects of the model.

The interaction grid of the model is available to download (S2 File) and is summarised in Fig 2.

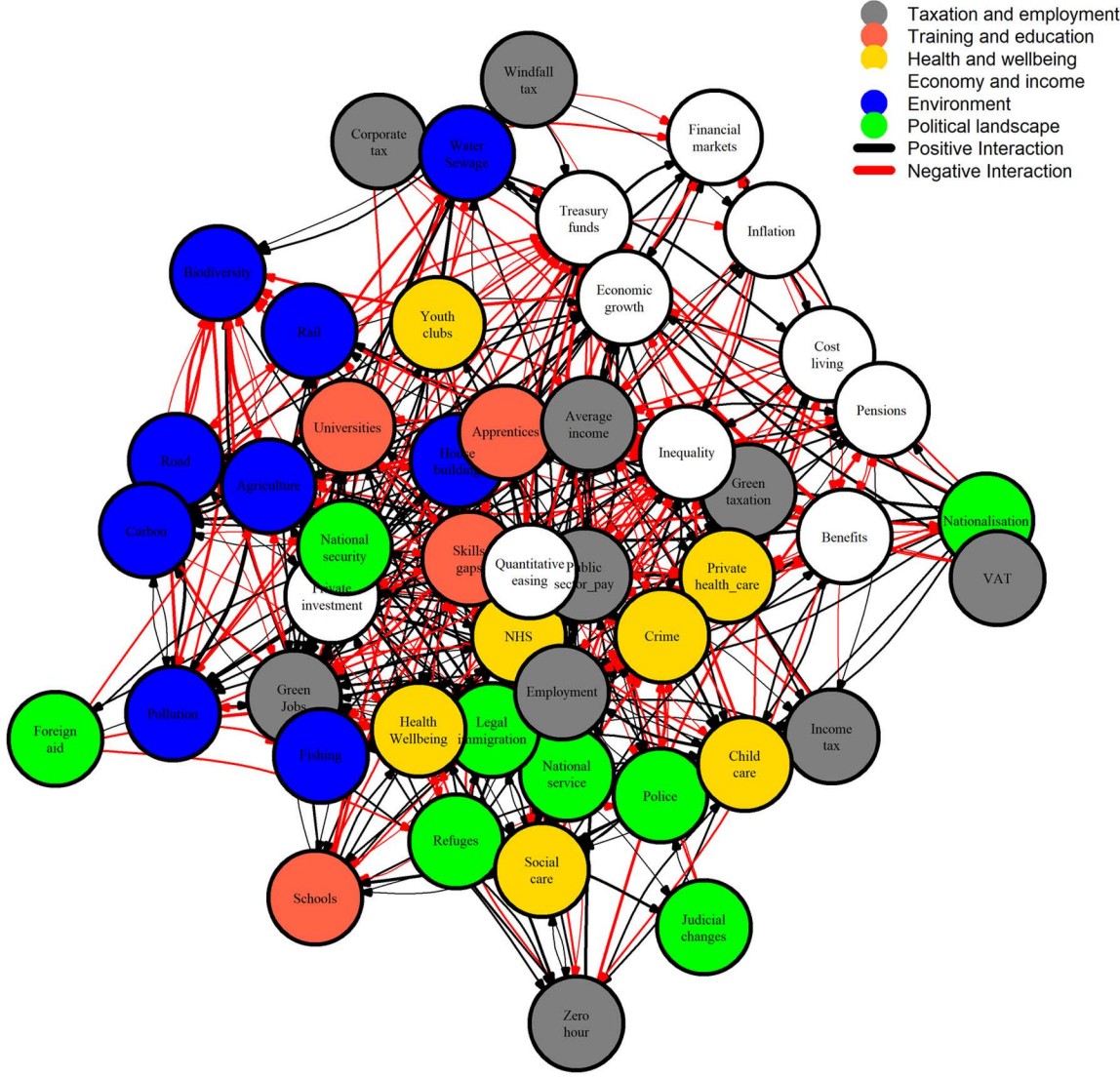

**Fig 2. Interaction diagram of the Bayesian belief network, demonstrating complex interactions between nodes (positive interaction: black arrows; negative interaction: red arrows) and between different categories of node (Taxation and employment: grey, Training and education: tomato red; Health and wellbeing: gold; Economy and income: white; Environment: blue; Political landscape: green).**

## 2.2. Node selection

To establish the nodes for the model, political manifestos were scrutinised to establish key policies (things a government can directly control) and outcomes (environmental, economic and societal outcomes which are influenced by one or more policies). Many nodes (such as economic growth) were desired outcomes of many of the political parties, while others, such as national service or community youth clubs were specific to just one of the parties. Nodes were also selected if they had demonstrated importance in previous work or could be assumed to be good societal indicators. For example, green taxation was shown to be important in environmental policies [18,19], and socio-economic inequality, although difficult to define in conventional units, is a good measure of income distribution in a society [26]. Finalisation of the nodes

was agreed in joint meetings of the authors. Together, the authors have collective expertise in complex systems models, political science, policy, environmental science and sustainability, political and human geography, and health, and where expertise was missing, independent experts and practitioners were invited to participate (e.g., immigration law).

### 2.3. Defining edges and edge strengths

Direct cause and effect interrelationships are represented in the model by positive or negative values (edges) that have estimated values between 1 and 4. The value 1 indicates a weak relationship, and the value 4 indicates a strong relationship. These values arise, where possible, from well-established evidence that directly links policies or services and outcomes (or directly links different nodes in the network). Positive values are recorded where: as one policy or service increases - the outcome also increases (this is a positive edge). Negative values are shown where a policy or service node increases and the affected node decreases. Only direct connections between nodes are modelled. For example, greater employment levels created a greater amount of income tax (across the UK population), but did not directly affect Treasury funds within the model, as this was an indirect ling, via the income tax node.

Edges and edge strengths were determined through individual analyses of the nodes followed by collaborative discussions involving the paper's authors to assess intercoder reliability (see also section 2.2 for details of author discussions).

Following individual scoring and group discussion of the edges and edge strengths, a sensitivity analysis on edge strengths was conducted using the BBNet package in R [27]; the 40 edge strengths (~ 10% of the total) with the highest cumulative impact on the nodes 'Crime', 'Carbon', 'Inflation', and 'Illegal immigration' (representing environmental, economic and social outcomes) were checked against academic literature to ensure their values were accurate (see sensitivity analysis procedure in [21] for full details). All but one of these 40 nodes remained unaltered after justification by the literature, illustrating that the group discussions had created a robust model (S1 File).

### 2.4. Creating scenarios based on political party manifestos

Scenarios are the mathematical representation of the political party manifestos as a series of scores (or prior values) for nodes which are represented by manifesto policies. The scenarios were created from the political manifestos of major political parties operating across all of England (Conservatives [12], Labour [13], Liberal Democrats [28], Green [29], Reform UK [11]). While many of these manifestos also apply to the UK's devolved nations, we did not consider devolved nation only parties in our analysis due to the complexity of comparing results across a larger number of parties.

The political party node interactions were defined in strength and direction by examining manifesto pledges on direct action and clarity of detail explicit in the manifesto. For example, a pledge to better fund the NHS with no further information may result in a value of +1 being given for the NHS node in the scenario. Whereas a commitment to improve annual funding by £28 billion (~17% of the total budget, as described by the Green Party [29]) would be given a value of +4. However, nodes are only changed with direct targeted action. For example, a statement reading: '£10 billion extra per annum will be provided to the NHS to improve health and well-being of the UK population' would only result in changes to the NHS node (note, this is an example quote, to clarify the point, and it is not present in any of the manifestos). The BBN modelling process would be used to determine changes to health and well-being as a result of the changes to the NHS. Scores were assigned as per the recommendations outlined in [21], with higher scores relating to higher certainty of the action and identified or higher magnitude of the action (e.g., providing more money). The scores for each scenario along with a full definition of what each node represents are shown in Table 1. Justifications of the scores from the manifestos are also provided (S3 File).

Unlike the majority of models used with the BBNet package, many of the model nodes in the current study were both input and output nodes (i.e., they would form the basis of prior values, but these prior values should also change to reflect positive and negative feedback caused by interactions between nodes). Consequently, nodes for each of the five political party manifestos were added to the model post sensitivity analysis (see S2 File), and for example, to examine the Conservative party manifesto, a prior was set of 'Conservative' to value +4 with all other node values in this scenario assigned prior values of 0 (further details provided in [21]).

**Table 1. Nodes used in the model, with definitions applied and 'prior' values which formed the edge strengths for each political party node.**

| Node | Definition | Green | Reform UK | Liberal Democrat | Conser-vative | Labour |
|---|---|---|---|---|---|---|
| *Taxation and Employment* | | | | | | |
| Income Taxation | Total income from all taxation on wages/pensions etc including national insurance | 3 | -2 | -2 | -1 | 1 |
| Green Taxation | Taxation on polluting things such as fuels (i.e., coal), or on products which damage biodiversity | 2 | -3 | | -1 | |
| Corporate Taxation | Taxation on large businesses profits | | -2 | 2 | | -1 |
| Windfall Taxation | Specific tax on fossil fuel companies - or other highly harmful environmental industries | 2 | | 2 | 1 | 3 |
| VAT | Amount of VAT paid | -1 | -2 | | -1 | |
| Employment | Number of people in full-time or part-time employment | | 1 | | | 1 |
| Public Sector Pay | Average pay in public sector work | | -1 | | | |
| Green Jobs | Jobs in green industries, i.e., working with nature, retrofitting houses, producing renewable energy | 3 | -3 | | | 3 |
| Zero Hour | Proportion of employment on 0 hours contracts | -1 | 2 | -2 | | -4 |
| Average Income | Mean income across working population (after taxation) | 2 | | | 1 | 1 |
| *Training and Education* | | | | | | |
| Apprentices | Increasing apprenticeship schemes | 2 | 2 | 3 | 2 | 3 |
| Schools | Performance of schools - more money will increase performance, as would fewer students or more teachers etc | 1 | 0 | 2 | 1 | 2 |
| Universities | Performance and funding of universities through teaching and research | 2 | -3 | 2 | -2 | 1 |
| Skills Gaps | Lack of supply for certain skilled professions | | -1 | | -2 | |
| *Health and Wellbeing* | | | | | | |
| NHS | Performance and funding of NHS | 4 | 1 | 2 | 2 | 1 |
| Health and Wellbeing | A general 'wellness' index, based on physical and mental health | 2 | -1 | 2 | | 2 |
| Social Care | Performance and investment in social care | 4 | -1 | 2 | 2 | 1 |
| Youth Clubs | Availability of youth clubs and similar schemes | | | | | 2 |
| Private Health Care | Health care not provided through the NHS | | 3 | -1 | | 1 |
| Child Care | Provision and cost of childcare | 4 | 0 | 1 | 3 | 2 |
| Crime | Illegal activity | -1 | -2 | | -1 | -2 |
| *Economy and Income* | | | | | | |
| Economic Growth | Increase in Gross Domestic Product (GDP) | | | | | 1 |
| Inflation | General increase in the cost of buying things; a change in the purchasing value of money | | | | | |
| Financial Markets | Confidence of financial markets | -3 | | | | 2 |
| Inequality | Discrepancy in wealth between richest and poorest | -2 | 1 | | | -2 |
| Quantitative Easing | Printing money to add to UK economy | 2 | -2 | | | |

*(Continued)*

**Table 1.** (Continued)

| Node | Definition | Green | Reform UK | Liberal Democrat | Conser-vative | Labour |
|---|---|---|---|---|---|---|
| Private Investment | Investment from non-government sources | 2 | 2 | 2 | 1 | 4 |
| Pensions | Average pension amount (after taxation - also see average income) | | | 3 | 2 | 1 |
| Benefits | All benefits, including disability benefits, universal credit etc. | 3 | -2 | 2 | -2 | |
| Cost of Living | General costs of goods and ability to afford them | | -1 | | | -2 |
| Treasury Funds | Money in from tax vs money out from expenditure | | 1 | | 1 | 1 |
| *Environment* | | | | | | |
| Carbon | A net carbon figure for UK activities | -3 | 4 | -3 | 1 | -2 |
| Biodiversity | Overall biodiversity and nature levels in the UK | 3 | -3 | 4 | | 1 |
| Pollution | Pollution of rivers, land, air and seas | -2 | 2 | -3 | 3 | |
| Agriculture | Amount of agriculture in the UK - note, works on traditional agricultural practices - 'green' agriculture should also result in changes to other nodes | -1 | 2 | 1 | 2 | 1 |
| Fishing | Amount of fish and seafood caught in the UK | -2 | 2 | -1 | 1 | |
| House Building | Schemes to increase housing, including affordable housing | 3 | 2 | 4 | 4 | 4 |
| Road | Investment and use of roads. Note - busses likely to reduce overall road use | -3 | 3 | -2 | 3 | 3 |
| Rail | As above, but with railways and trains | 3 | -1 | 3 | 1 | 2 |
| Water and Sewage | Effective functioning of water companies | 3 | | 4 | | 1 |
| *Political Landscape* | | | | | | |
| Nationalisa-tion | Public ownership of some utilities | 3 | | 1 | | 2 |
| Judicial Changes | Changes to legislation especially around classifications of crimes and prosecution | -1 | 3 | | 2 | 3 |
| Police | Performance and size of the police force | | 3 | 1 | 1 | 2 |
| National Security | Security from international risks, including size and investment in military | | 2 | 2 | 2 | 2 |
| Refuges | Refugees and asylum seekers - non-regulated entry to country | 2 | -2 | 1 | -1 | -2 |
| Legal Immigration | Regulated entry to country | 3 | -2 | 2 | -2 | -2 |
| Foreign Aid | Total money provided for foreign aid | 3 | -3 | 1 | | 1 |
| National Service | Conservative scheme for military and social volunteering | | | | 2 | |

## 2.5. Model and parameter subjectivity

Models are a simplified version of the real world and are subject to error in terms of structure and parameter estima-tion. The BBNet package used here allows for complex systems to be modelled (i.e., a series of interacting nodes) but simplifies the direct connections between the nodes. The 1–4 scale used to assess interaction strength is designed to be used with collective opinion, or to be easily assigned from literature (e.g., a strong effect reported in many studies should be scored 4 – see details in [21]). As such, collective agreement on scores within a large multidisciplinary group of researchers is possible (see section 2.3). The sensitivity analysis also assesses and ranks the edge strengths most likely to affect the key results, so these can be checked further against literature, where available. Modelling of only the

direct interactions also helps avoid wrongly assigning complex non-direct interactions when assigning edges in the model. In these models, edges and edge strength are static (i.e., they do not change depending on the model conditions, even though the node values do). Where this may be unrealistic (e.g., identifying the effects of increasing debt on financial market behaviour, which can be difficult to predict), we have tried to eliminate the need for the model to change or adapt. For example, our values for financial markets and treasury funds nodes in the model are treated as outputs, rather than trying to model their effects on the rest of the system (see section 2.1). The simplicity of the modelling approach allows for more complex models to be made. However, the precision of the outputs is not necessarily high, or in meaningful units. For example, we can not predict the deficit of treasury funds in pounds sterling, but we can provide ordinal responses to which party's policies will provide the largest deficit, and this level of precision is reflected in the results.

## 2.6. Running the models

Models were run using the BBNet package [27] on R version 4.3.2 [30]. The bbn.predict function was used to collate posterior values for each of the nodes. Values for all nodes were ranked by political party before presenting, as per [19]. Ranks ranged between 1 and 5, with 5 representing the party with the 'best' outcome for that particular node. The 'best' outcome is clearly subjective, and our ranking is based on what is assumed to be a popular outcome. For example, most voters would welcome lower taxation, but would also rather a better performing NHS. More controversial ranking, such as that for immigration, is discussed below, but we assumed lower immigration would be a more favourable outcome for the general public.

## 2.7. Assessing the importance of the modelling in visualising future political landscapes

Multiple opinions and evaluations of manifestos are written by academics and journalists during election campaigns, sometimes focussed on particular policy areas [31,32]. As such, we wanted to determine additional insights or changes in insights which could be gained by creating predictive models. We examined the number of nodes we had no information on (i.e., scored zero) both in our scenarios and our final predictions. We also performed cluster analysis (using the heatmap function in R, using the default hclust clustering formula) to examine similarities between political parties both on 'prior' information from the scenarios and the 'posterior' information, following the running of the models to demonstrate the spatial positioning of parties, and how this positioning changes with new information from the modelling process.

## 3. Results

While a number of policies were included in the scenarios for each manifesto, and given initial scores, the model generated considerably more information for nodes not directly mentioned in the manifestos and altered the values of many of the nodes which were initially scored in the scenarios. Creating the input scenarios for model resulted in 70 nodes not being given values across all five parties (mean of 14 out of 48 nodes for each manifesto – Table 1). Following the modelling process only 13 nodes (mean of 2.6 per party) were scored as 0 (had no information), creating a more complete picture of what may happen to certain aspects of society if a party came to power and enacted their manifesto in full. For example, the Liberal Democrats did not make explicit commitments to green jobs in their manifesto, yet our model predictions resulted in the Liberal Democrats creating the highest number of green jobs of all parties, through the provision of other strong policies, particularly on the environment. In addition, prior to modelling scenarios, cluster analysis predicted that Reform and the Conservatives had closely related policies, with the Liberal Democrats and Greens also being similar. Labour was aligned, although less strongly, with the Greens and Liberal Democrats. Following the modelling process, Reform was a clear outlier in the analysis, with Labour and Conservatives clustering closely, and the Greens and Liberal Democrats still clustering closely (Fig 3).

To identify which party generates the best (ranked score of 5) and worst outcomes (ranked score of 1) from their proposed policies, the ranked values from each of the 48 nodes were counted and presented as histograms (Fig 4). Nodes for the Green Party manifesto and Reform UK's manifesto showed high counts of nodes ranked as 1 or 5. Table 2 presents the ranking 'order' for each node, generally considered as 'positive' outcomes for each node.

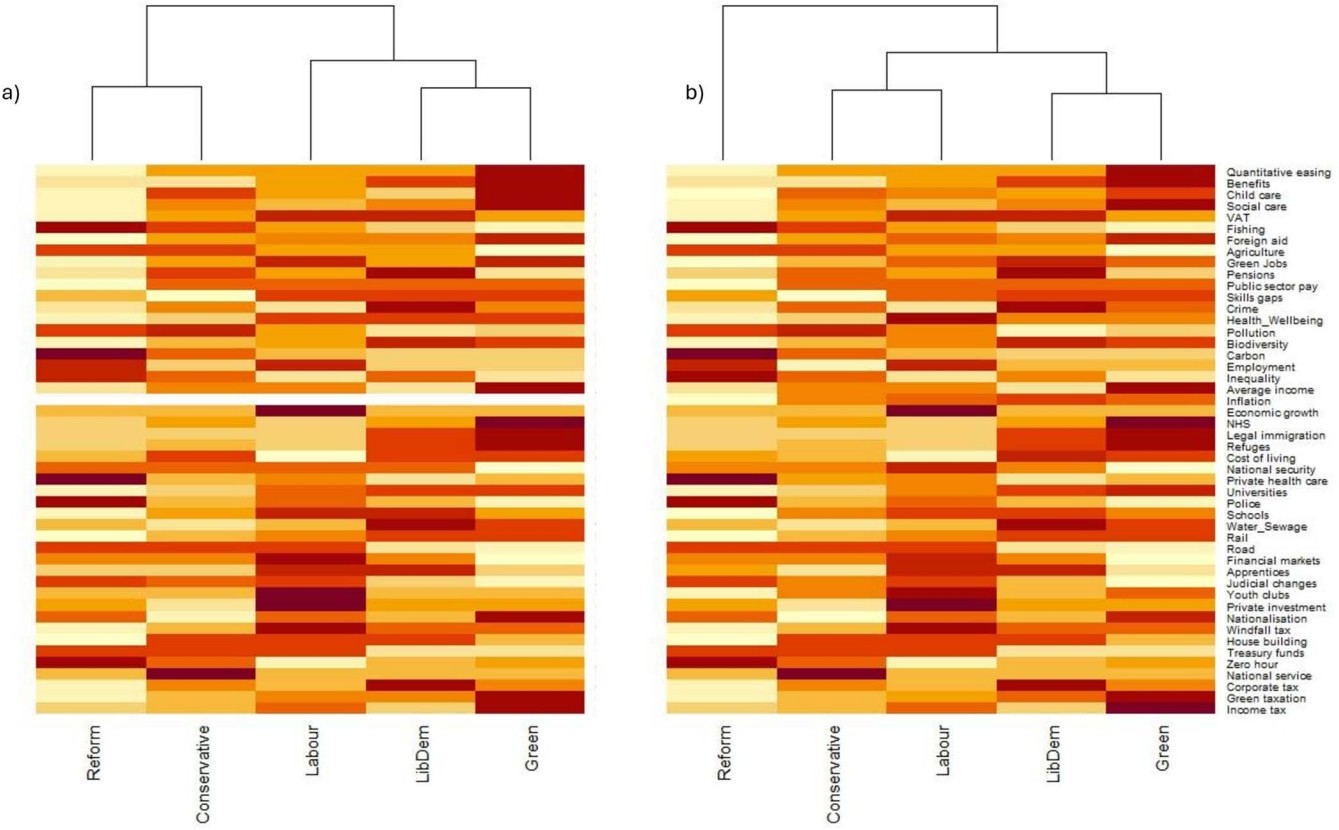

**Fig 3. Heatmaps of how the different manifestos compare across the different network model nodes as (a) 'prior' values (data taken solely from manifestos for each node) and (b) 'posterior' values (calculated values from the network, indicating complex relationships between nodes).** Dendrograms show cluster analysis and relatedness of different political party manifestos.

An important caveat is that this ranking order is highly subjective. For example, the Green Party are clear that they do not wish to limit numbers of refugees, asylum seekers, and legal migrants. While their policies result in the worst performance to 'reduce immigration' of all the political party manifestos, this is very much a value judgement on migrants, and the score of 1 here needs to be interpreted in context. Nevertheless, the larger number of posterior nodes ranked as 1 and 5 for Green's and Reform (and to some extent the large number of 2 and 4 scores for the Liberal Democrats) do indicate greater divergence of policies from those of the other parties which can be thought to 'occupy the middle ground'. The Labour histogram is slightly skewed towards higher rankings compared to the Conservative histogram, and as such, it is likely the Labour manifesto provides policies which would resonate more with the general public than the Conservative manifesto (again, subject to the value judgements provided for the order of ranking provided in Table 2).

### 3.1. Green Party manifesto outcomes

The Green Party manifesto provides policies with strong outcomes for the environment (Table 2, see also section 3.1). While it is low ranked for tax reduction (in most cases, taxes would increase) it also provides the highest average income and public sector pay, scores well for reducing economic inequality, increasing childcare and benefits, clearly demonstrating second order consequences of taxation. While not an explicit priority of the manifesto, it is predicted to deliver the second highest level of economic growth of all the political parties, an emergent result of other policies. Public services

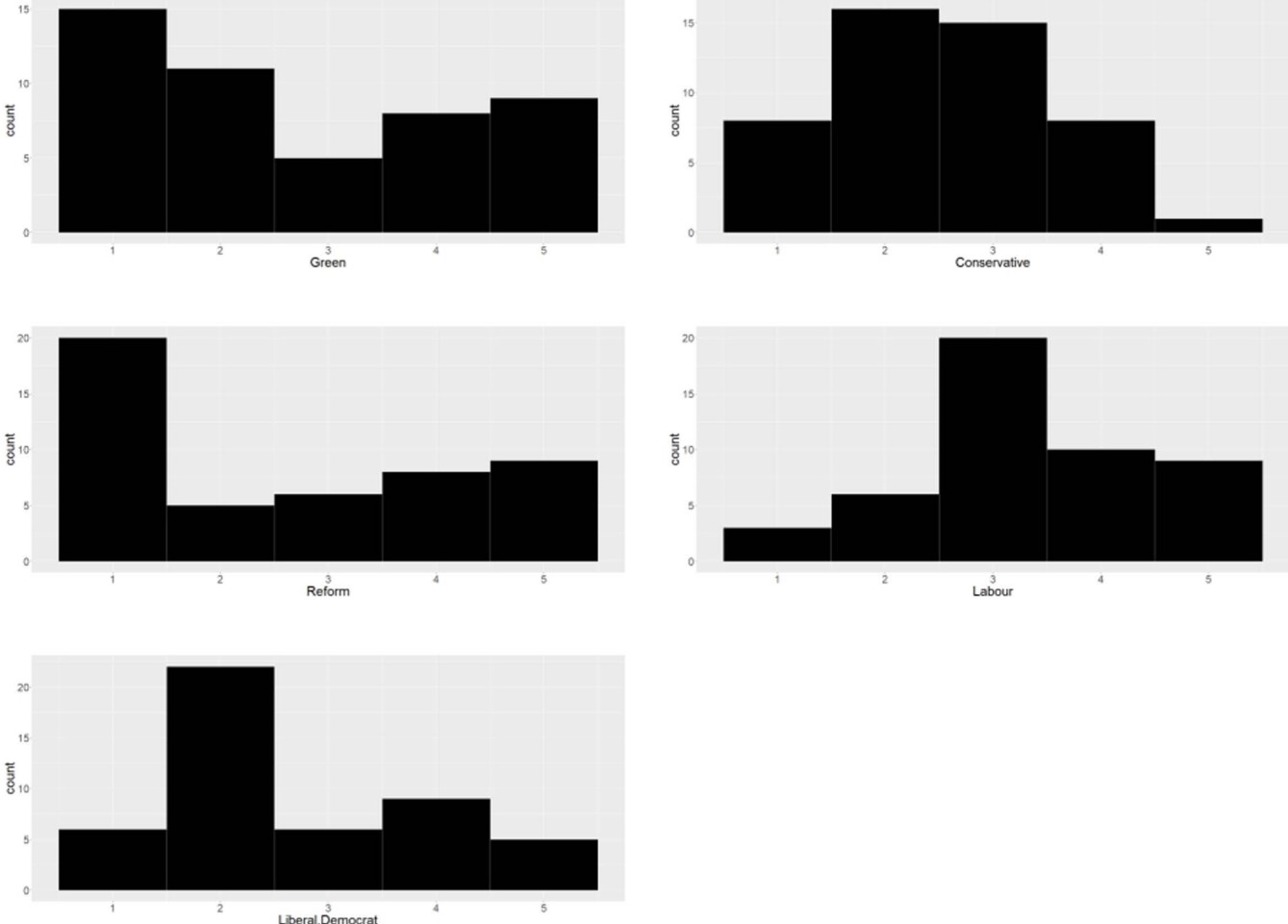

**Fig 4. Histograms of ranking score for each manifesto across all 48 model nodes.** Ranking of 5 is based on the manifesto which provides the strongest response (based on the 'direction' indicated in Table 2). For example, a score of 5 represents the manifesto which we predict will result in the strongest predicted economic growth or the lowest level of legal migration. Note, y axis range differs between political parties.

also perform well, with the best outcomes for the NHS, rail and water and sewage. The manifesto policies are predicted to perform poorly on crime and national security, reducing immigration (see also section 3.3) and food production (agriculture and fishing). Due to regulation and likely increased financial deficits from proposed Green policies, in our model, the financial markets are predicted to be nervous of these policies, and problems from financial market instability can be catastrophic economically and politically, for example, as indicated by the rapid demise of the Liz Truss government in 2022 [33]. While it is unlikely that the Green Party will be elected to government, the potential concerns of the financial institutions do need to be considered.

### 3.2. Reform UK manifesto (contract) outcomes

Largely predictions from this manifesto are the opposite of those for the Green Party. Reform perform very poorly on the environment. Tax cuts are typically the highest of any party, but nevertheless, average income, public sector pay and economic equality are the lowest values for any party, and are predicted to get worse than current levels, again illustrating

**Table 2.** Modelled policy outcomes, with posterior values for individual nodes ranked from 1-5 across the parties - Green, Reform UK, Liberal Democrat, Conservative, and Labour, where 1 is the lowest rank and 5 is the highest. The parties are ranked for each policy node in the network (e.g., income tax), based on the direction of change ("performance indicator") stated in the first column (e.g., lowest, highest, most, improvement). For example, a value of 5 for Green tax means that green tax is lowest under Reform UK's policies, while a value of 1 means that green tax is highest under the Green Party's policies. Larger text in bold and italic font indicates that the directional change (i.e., increase or decrease) of a node under a party's policies is consistent with the direction indicated, whereas black text indicates an opposite directional change (e.g., income tax is predicted to decrease under Reform UK in alignment with the performance indicator, and increase under Green and Labour).

| Taxation and employment | Green | Reform UK | Liberal Democrat | Conservative | Labour |
|---|---|---|---|---|---|
| Income tax - lowest | 1 | *4* | *4* | 3 | 2 |
| Green tax - lowest | 1 | *5* | 2 | *4* | *3* |
| Corporate tax - lowest | 2 | *5* | 1 | 2 | *4* |
| Windfall tax - lowest | 2 | 5 | 2 | 4 | 1 |
| VAT - lowest | *3* | *5* | 1 | *3* | 1 |
| Employment - highest | *3* | *4* | *2* | 1 | *4* |
| Public sector pay - highest | *5* | 1 | *4* | 2 | 3 |
| Green Jobs - most | 3 | 1 | *5* | *2* | 3 |
| Zero hour - lowest | 3 | 1 | *4* | 2 | *5* |
| Average income - highest | *5* | *1* | 2 | 3 | 3 |
| | | | | | |
| **Training and education** | Green | Reform | Liberal Democrat | Conservative | Labour |
| Apprentices - highest | *1* | *3* | 4 | 1 | *5* |
| Schools - improvement | *2* | *1* | 4 | 2 | *4* |
| Universities - improvement | *5* | 1 | 4 | 2 | 3 |
| Skills gaps - smallest | *1* | *4* | 2 | *5* | 3 |
| | | | | | |
| **Health and Wellbeing** | Green | Reform | Liberal Democrat | Conservative | Labour |
| NHS - best performing | *5* | 1 | 3 | 3 | *1* |
| Health&Wellbeing - highest | *3* | 1 | 3 | 2 | *5* |
| Social care - highest | *5* | 1 | 3 | 3 | 2 |
| Youth clubs - highest | *4* | *1* | 2 | 3 | *5* |
| Private health care - increased | 2 | *5* | 1 | 3 | 4 |
| Child care - highest | *5* | *1* | 2 | 4 | 3 |
| Crime -lowest | *2* | *4* | *1* | 2 | *4* |
| | | | | | |
| **Economy and income** | Green | Reform | Liberal Democrat | Conservative | Labour |
| Economic growth -highest | *4* | *1* | *2* | 3 | *5* |
| Inflation - lowest | 2 | *5* | 1 | 4 | 3 |
| Financial markets - securest | 1 | *5* | 2 | *3* | *4* |
| Inequality -lowest | *4* | 1 | *3* | 2 | *4* |
| Quantitative easing - lowest | 1 | *5* | 2 | 2 | 2 |
| Private investment - highest | *2* | *2* | *2* | 1 | *5* |
| Pensions - highest | *2* | 1 | *5* | 4 | 3 |
| Benefits - highest | *5* | 1 | *4* | 1 | 3 |
| Cost of living - lowest | *2* | *3* | 1 | *4* | *5* |
| Treasury funds - highest | 1 | *3* | 2 | 3 | 3 |
| | | | | | |
| **Environment** | Green | Reform | Liberal Democrat | Conservative | Labour |

*(Continued)*

**Table 2.** (Continued)

| Taxation and employment | Green | Reform UK | Liberal Democrat | Conservative | Labour |
|---|---|---|---|---|---|
| Carbon - lowest | *4* | 1 | *4* | 2 | *3* |
| Biodiversity -highest | *4* | 1 | *5* | 2 | *3* |
| Pollution - lowest | *4* | 2 | *5* | 1 | 3 |
| Agriculture - highest | 1 | *4* | *2* | *4* | *2* |
| Fishing - highest | 1 | *5* | 2 | *4* | 3 |
| House building - highest | *2* | *1* | 3 | *3* | 3 |
| Road - highest | 1 | *3* | 2 | *3* | 3 |
| Rail - highest | *4* | 1 | *4* | 2 | *3* |
| Water&Sewage - improved | *4* | 1 | *5* | 2 | *2* |
| | | | | | |
| **Political landscape** | Green | Reform | Liberal Democrat | Conservative | Labour |
| Nationalisation - highest | *5* | 1 | *3* | 2 | *4* |
| Judicial changes - most restrictive | 1 | *4* | *2* | 3 | *4* |
| Police - increased capacity | 1 | *5* | *2* | 2 | *4* |
| National security - highest | *1* | *2* | *2* | 2 | *5* |
| Refuges - lowest | 1 | *4* | 2 | 3 | *4* |
| Legal immigration -lowest | 1 | *3* | 2 | 3 | 3 |
| Foreign aid - highest | *5* | 1 | *3* | 2 | *4* |
| National Service - highest | *1* | *1* | *1* | *5* | *1* |

the second order consequences of taxation (Table 2). Schools, universities, the NHS and social care are also predicted to perform poorly under the Reform policies. The model also predicts that economic growth is also likely to be lowest under Reform policies, compared to the second highest for the Green Party. Reform policies support the development of agriculture and fisheries (although from a perspective of increased supply to the UK, not in terms of longer-term sustainability of the industries). It performs well in reducing crime rate and it is only joint strongest in reducing both legal immigration (tied with Labour and Conservatives) and refugees and asylum seekers (tied with Labour).

### 3.3. Liberal Democrat manifesto outcomes

In many cases, Liberal Democrat policies result in similar outcomes to the Green Party. They perform very well on environmental improvements, as well as services such as rail and utilities. They are strong on training and education (schools and universities). The model predicts weaker outcomes for crime and reducing immigration than many other parties. Many taxes would likely increase, although revenue from income tax would likely decrease with increased payment thresholds proposed. However, these do not correspond quite as strongly as Green policies to increases in average wages and economic inequality. Along with Labour and Conservative pledges, there is a very strong commitment to house building, which for the Liberal Democrats, does not greatly diminish their environmental credentials.

### 3.4. Conservative manifesto outcomes

Economically, the Conservative picture largely follows that of Reform, with lower taxes, but lower average incomes and public sector pay than most other parties, although the rankings are less extreme than Reform. Our model suggests it would deliver poorly on employment, increases to private investment, inflation and crime, traditionally areas associated with the Conservative party [34]. While changes are likely to be modest, the Conservatives are predicted to perform better on the NHS than Labour, and well on pensions and the cost of living. It performs poorly on the environment and

is predicted to provide the worse outcomes for pollution of all the parties (from outputs of both models), with very little to address current controversies such as sewage in rivers. The Conservative policies only result in one ranking of 5 across all of the nodes, this being for National Service, of which it is the only party to propose this policy.

### 3.5. Labour manifesto outcomes

The Labour manifesto was the longest of those studied, yet in many places had little detail and specifics of their policies. Perhaps because of this, Labour also ranked 'mid-table' for many predicted outcomes, although as indicated in section 3.3, normally a little higher than Conservatives. Labour policies were predicted to perform well in areas not really associated with Labour, for example, their policies were predicted to cause the strongest economic growth, some of the most secure financial markets, lowest cost of living and the highest amount of private investment into the economy. However, Labour performed poorly in areas where they may be thought of as traditionally strong [35]. These include the joint worst outcomes for the NHS and the second worst outcomes for social care (although these were predicted to improve, rather than get worse). House building, reducing legal migration and reduction of refugees and asylum seekers were also strong under Labour, as was overall employment, apprenticeships, and a commitment to end zero hours contracts. From an environmental perspective, Labour ranked in the middle of the five parties

## 4. Discussion

Our results provide clear evidence to support the aims of the study. For example, the extra information obtained from the models, in terms of the number of nodes not scored as zero, indicates the modelling process provides a more holistic picture of the environmental, societal, economic and political landscape than manifestos do alone. This extra information also changes the spatial positioning of the parties compared to an examination of policies from the manifestos alone and allows us to paint a clearer picture of the key outcomes a party manifesto would achieve, if enacted in full. We also clearly identify unintended, or unreported, consequences of policies (or second order effects, not clearly described as outcomes within manifestos).

### 4.1. Holistic outcomes of manifestos in achieving societal, environmental and economic outcomes

Our work has produced a synthesis of likely outcomes to many economic, environmental, and social aspects of pledges in political party manifestos. The two largest parties (Labour and Conservative) have more mid-ranking policies than the smaller parties and are relatively similar to each other in terms of outcomes, despite traditionally having different values [34,35]. Generally, of the two, Labour are predicted to provide better outcomes (in relation to alignment with common viewpoints on what would be optimal outcomes) than the Conservatives.

The Green Party and Liberal Democrats provide much stronger environmental outcomes than the two major parties and are predicted to provide better public services. The Green Party also outperforms others in terms of addressing economic inequality and is the second strongest party in achieving economic growth. While this may seem counter to the degrowth agenda of many Greens, the unintended or agnostic response to economic growth does fit within the concept of Doughnut Economics, often embraced by the Greens [36].

Poor performance of the Liberal Democrats and Greens on immigration does reflect what is said in their manifestos and needs to be considered in light of our ranking of immigration being perceived negatively by the public, when clearly it is a divisive issue [37]. Poor performance in areas such as fishing and farming are also based over a five-year parliamentary time frame. Considerable commitment is made to 'greening' these industries, and long-term benefits may arise from more sustainable approaches (e.g., [38,39]).

Reform, and to a lesser extent the Conservatives, perform very poorly on the environment and on public services. While taxes are reduced, average incomes and inequality can be poor. Crime (under a traditional model of more enforcement and higher penalties leading to reduction of crime rate) is a strength of the Reform policies. While a contentious

issue, migration (legal and otherwise) is predicted to increase under Green and Liberal Democrat policies, but modestly decrease under the other parties' policies (notably Reform does not perform better than Labour, despite the rhetoric used by the party). Of particular interest is the poor economic growth performance predicted from Reform, perhaps demonstrating that claims of the fossil fuel lobby that these products are vital to retain high growth rates are unfounded [40].

Typically, many pledges within manifestos will go unfulfilled over the course of a five-year parliament [6,7,9]. Financial instability can be a significant reason for this unfulfillment, with financial crises changing political priorities (e.g., the 2008 financial crash [41] and the financial institutions' response to the Liz Truss government's budget [33] being two examples). Since financial institutions' decisions can be highly unpredictable, in this study, we highlight the likely security of the financial markets to income generation and spending plans. Liberal Democrats and Greens fair most poorly in this regard, potentially due to high spending plans, even if somewhat funded by tax rises. Such instability may limit the ability of governments to achieve their aims, as a focus may need to be placed on financial stability [33]. In contrast, Labour, traditionally labelled as poor on the economy [42], demonstrate the highest values of financial stability, potentially overcompensating to address criticism from others in this area.

## 4.2. Unintended or second order consequences of policies

Our analysis identifies several indirect effects of policies, which are either unintended or counter to the general impressions provided within manifestos. Perhaps the biggest influence is effect of personal tax cuts (e.g., income tax, green taxation and VAT) as proposed by Reform and the Conservatives. While such tax cuts are presented as a method to maximise personal income, we show the parties offering the biggest tax cuts also resulted in low levels of average income (which included money from benefits, pensions etc) and higher levels of socio-economic inequality. While the concept of 'the rich get richer' has been associated with more conservative policies (both in the UK and US) for some time [43], tax cutting policies are targeted at all working people. With the majority of reform voters thought to come from a poorer demographic [44], it is certainly interesting that they are unlikely to realise the policies they are voting for will make them worse off, with our analysis showing that policies on taxation not benefiting the majority of working people, but rather the richest in society.

Perhaps surprisingly, the Liberal Democrats were the party with the best predicted environmental outcomes overall, performing slightly better than the Greens. As discussed above (section 4.1), Green policies resulted predictions of high economic growth. This increase in economic growth may limit some environmental benefits. While the debate on decoupling economic growth from environmental degradation is still ongoing, most research suggests that this can only be partially achieved at best [45]. This may partly explain why the Greens did not perform as well on issues such as climate change as the Liberal Democrats.

Another surprising result came from the predictions of the parties to immigration. Reform and Conservatives had a very strong rhetoric on immigration prior to the election, yet we predicted marginally better outcomes for reducing immigration (legal and illegal) from Labour, compared to Conservatives, and Labour performing equally to Reform. This may be due to strong policies from Labour on employment, apprenticeships and zero-hour contracts which will make the UK workforce more able to compete with immigrant labour [46]. However, while multiple policies can also affect crime rates, we demonstrate that strong policing, as proposed by Reform still had a strong outcome on crime rates, although again scoring equal to Labour, with slightly less emphasis on police, but with additional policies to reduce inequality.

## 4.3. Spatial positioning of the political parties

Considerable research has been conducted on spatial positioning of political parties on right to left spectra, or more complex multidimensional plots [47]. Understanding where political parties are spatially located can help them design policies or suggest outcomes in manifestos to compete for voter share with other parties. For example, in general, over the last 30 years, major UK parties have occupied more central ground than they may have prior to this [48].

Although not using a traditional left to right framework, our analysis demonstrates important differences in positioning of parties based on an analysis of policies (our scenarios for the models), compared to an analysis of policies and outcomes (our modelled predictions). Based on policies alone, Conservative and Reform showed similar positioning, with Labour, Liberal Democrats and Greens demonstrating a separate cluster (Labour more different than the other two parties in this cluster). However, based on model results, Labour and Conservatives showed similar model outcomes (the centre ground), with Reform showing as a clear outlier. Equally, our histogram analysis showed both Labour and Conservative showing the majority of mid ranked policies, essentially trying to appeal to some extent to all voters on all policies, rather than the more polarised responses of other parties, providing more robust responses to particular outcomes.

Reform becoming an outlier from the other parties in terms of its outcomes, is of particular importance. Reform is frequently seen as a 'populist' party, and also often referred to in the media as a 'far right' party. Such parties often try to court public opinion on topics of concern, in order to pursue their own agendas [49]. Accordingly, individual policies did not appear to differ greatly from more mainstream right of centre parties such as the Conservatives, but the overall outcomes of their policies resulted in the party being an outlier (see also section 4.2 with respect to the effects on inequality and average wages).

While our analysis does not align to the traditional methods to position parties, the analysis of results does demonstrate alignment of outcomes, or centre ground, of the two main parties, and the real potential of change of political landscape belonging to the smaller parties, although it is unlikely that these parties would be able to achieve sufficient support to form a government in their own right.

## 4.4. Limitations and future uses of the model predictions

The models used, examine commitments in party manifestos, assign scores to these commitments, and investigate the complex interaction of different aspects of the economy, environment and social landscapes to provide a prediction. The biggest limitation is that manifesto pledges may be modified, refined or not undertaken, especially if a single party does not get an outright majority and needs to form a coalition government with others [50]; although in the case of the UK General Election in 2024, Labour have a very strong majority, and a strong mandate to deliver their manifesto in full. In addition, policies, especially tax rises, may be somewhat hidden within manifestos, with limited mention or detail, and therefore being subject to low scores in the model. Furthermore, the timeframe of the model being ~5 years (or one parliament) presents some limitations, such as the long-term sustainability of agriculture or fisheries are not fully addressed, nor is the longer-term issue of migration as climate change intensifies [51].

Predictions from models are only as good as the models themselves, which can be subject to error. While the simplicity of the modelling approach allows for very complex systems to be modelled (see section 2.5), inaccuracies can still exist. The simplification of the model from the real world could introduce errors. Furthermore, while the current model is an integrative approach to easily quantify or semi-quantify ideas (i.e., taxation and spending, carbon budgets, performance of large organisations), it is currently unable to capture and integrate more qualitative ideas. For example, education reforms proposed by proposed by parties such as Reform that repudiate transgender ideology and Critical Race theory in schools and online ("ban Transgender Ideology in Primary and Secondary Schools"; "social media giants that push baseless transgender ideology and divisive Critical Race theory should have no role in regulating free speech") around teaching that there are only two genders ("There are 2 sexes and 2 genders") or race ("Ban Critical Race Theory in Primary and Secondary Schools" – a quote only present in the draft manifesto, and removed in the final version) may have significant effects on some children's mental health, or long-term social equality and international [52,53]. Future research should address these shortcomings, as such qualitative information could be incorporated into models with sufficient empirical evidence to inform edge strengths.

While the model is currently based on the UK economy and social and environmental landscapes, models can be developed for other countries, or global landscapes. Once developed, these can easily be integrated with different

policies, not just during elections when large suites of policies are introduced, but to predict the effects of specific policy changes during the term of a government. They can also be used by think-tanks, NGOs and other policy influencers to evaluate potential policy ideas on a holistic set of outcomes.

Out model does use informed opinion to quantify or semi-quantify manifesto pledges and uses a pre-selected number of 'nodes' which need to be linked to details contained within manifestos or other policies. Future work can focus on linking our model approaches to other quantification processes, such as whether our predictions align with approaches such as sentiment analysis [54], and therefore reflect the intended messages in manifestos, or counter outcomes of this approach, thereby demonstrating the importance of second-order interactions which parties may be trying to hide from voters. It would also be useful to link the modelling process to analyses such as those produced by the Manifesto Project (e.g., [55]), which would allow rapid application of the technique to many manifestos from around the world. These manifestos include historic manifestos which would allow researchers to potentially examine the accuracy of the predictions over a five year period, or identify key policies resulting in significant changes to economic, environmental or social outcomes.

## 5. Conclusions

Journalists and commentators tend to concentrate on the presentation and launch of manifestos, the key takeaways and factcheck the claims made. They are an artefact of the campaign which highlights the priorities of the party as they court the voters for their support. From a reading of manifestos, it is difficult to see the effects of specific policies, and the outcomes ascribed to policy levers in manifestos are often taken at face value, rather than questioned. Policies can also have unintended (or intended but unmentioned) effects on other outcomes, so providing a holistic understanding of what a suite of policies will achieve is important in making voting decisions. It is this gap which this paper seeks to fill. By ranking each policy proposal as having an impact on other areas of policy we are able to identify the actual outcomes within key priority areas. Similar to voter advice applications, our research could be used by voters to assess how each party promises map onto their priorities. Voters asking the question 'do party promises mean actual change' can be guided somewhat by our research. Similarly, journalists could use these findings to assess the extent that broad party goals, such as moving towards a more sustainable economy or reducing poverty, will actually be realised. What our data also highlight is the difference between smaller parties who focus on specific goals and link their policy promises directly to those goals and catch-all parties. The latter, as proposed by Kirchheimer, attempts to balance competing demands to maximise voter satisfaction [56]. Hence, they will attempt to offer a little progress in multiple areas, which we propose was Labour's strategy in 2024. Hence our data offers a fresh perspective that can be utilised broadly to understand the interconnectedness of manifesto promises and the likely outcomes if manifestos were implemented in totality, recognising it may prove impossible and impractical for implementation in some cases and contexts. We hope the study provides valuable context for helping eliminate some of the discourse around the apparent polarisation of politics and associated rhetoric which has occurred over the last decade [57].

## Supporting information

**S1 File. Justifications of edge strength for the 40 most sensitive nodes of the Bayesian belief network.** Values before and after literature search are given, as are key supporting references.
(XLSX)

**S2 File. Full version of the model used in the analysis, including model interaction file, prior values used in the analysis and R code detailing sensitivity analysis and running of the model.**
(ZIP)

**S3 File.  Justification for values in different scenario nodes included in the models, based on statements made in different political party manifestos.**
(DOCX)

## Acknowledgments

We would like to thank Emma Tompkins (University of Southampton) for her wide-ranging insights and exceptional proof-reading - greatly improving the clarity of the manuscript, Rhiannon Croker (Duncan Lewis Public Law) for her detailed knowledge of UK immigration and seven anonymous referees whose comments greatly clarified and provided more context for this manuscript.

## Author contributions

**Conceptualization:** Richard Stafford, Jenny Alexander, Stephen Axon, Zach Boakes, Elena Cantarello, Abigail R. Croker, Marin Cvitanović, Victoria Dominguez Almela, Tilak Ginige, Darren Lilleker, Sarah J. Upson.

**Data curation:** Richard Stafford, Zach Boakes, Elena Cantarello, Abigail R. Croker, Victoria Dominguez Almela, Ellie-Anne Jones, Ellie Vincent.

**Formal analysis:** Richard Stafford, Stephen Axon, Zach Boakes, Elena Cantarello, Abigail R. Croker, Marin Cvitanović, Victoria Dominguez Almela, Tilak Ginige, Natalie Harris, Freya Russell, James R. Sokolnicki, Sarah J. Upson, Ellie Vincent.

**Investigation:** Richard Stafford, Jenny Alexander, Stephen Axon, Zach Boakes, Elena Cantarello, Marin Cvitanović, Victoria Dominguez Almela, Tilak Ginige, Natalie Harris, Ellie-Anne Jones, Yiannis Kountouris, Darren Lilleker, Freya Russell, James R. Sokolnicki, Sarah J. Upson.

**Methodology:** Richard Stafford, Abigail R. Croker, Victoria Dominguez Almela, Natalie Harris, Ellie Vincent.

**Project administration:** Richard Stafford.

**Resources:** Richard Stafford, Zach Boakes, Victoria Dominguez Almela, Ellie-Anne Jones.

**Software:** Richard Stafford, Victoria Dominguez Almela.

**Validation:** Richard Stafford, Elena Cantarello, Abigail R. Croker, Marin Cvitanović, Natalie Harris, Yiannis Kountouris, Freya Russell, Ellie Vincent.

**Visualization:** Richard Stafford, Victoria Dominguez Almela, Ellie-Anne Jones.

**Writing – original draft:** Richard Stafford, Jenny Alexander, Stephen Axon, Zach Boakes, Elena Cantarello, Abigail R. Croker, Marin Cvitanović, Victoria Dominguez Almela, Tilak Ginige, Natalie Harris, Ellie-Anne Jones, Yiannis Kountouris, Darren Lilleker, Freya Russell, James R. Sokolnicki, Sarah J. Upson.

**Writing – review & editing:** Richard Stafford, Jenny Alexander, Stephen Axon, Elena Cantarello, Abigail R. Croker, Marin Cvitanović, Victoria Dominguez Almela, Natalie Harris, Yiannis Kountouris, Darren Lilleker, James R. Sokolnicki, Sarah J. Upson.

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
