## [Decision Letter · Decision Letter 0]

8 Oct 2024

PONE-D-24-31129Evaluating post 2024 election scenarios for the UK based on political party manifestosPLOS ONE

Dear Dr. Stafford,

Thank you for submitting your manuscript to PLOS ONE. After careful consideration, we feel that it has merit but does not fully meet PLOS ONE’s publication criteria as it currently stands. Therefore, we invite you to submit a revised version of the manuscript that addresses the points raised during the review process.

While the reviewers find the paper to be an interesting and potentially significant contribution to the academic study of party politics, especially regarding manifestos, they also raise some concerns about the framing of the paper, engagement with literature, methodological clarity, and neglect of political context.

We look forward to receiving your revised manuscript.

Kind regards,

Ching-Hsing Wang, Ph.D.

Academic Editor

PLOS ONE

Journal Requirements:

Reviewers' comments:

Reviewer's Responses to Questions

**Comments to the Author**

1. Is the manuscript technically sound, and do the data support the conclusions?

Reviewer #1: No

Reviewer #2: Yes

Reviewer #3: No

Reviewer #4: Yes

2. Has the statistical analysis been performed appropriately and rigorously? 

Reviewer #1: No

Reviewer #2: Yes

Reviewer #3: No

Reviewer #4: Yes

3. Have the authors made all data underlying the findings in their manuscript fully available?

Reviewer #1: No

Reviewer #2: Yes

Reviewer #3: No

Reviewer #4: Yes

4. Is the manuscript presented in an intelligible fashion and written in standard English?

Reviewer #1: Yes

Reviewer #2: No

Reviewer #3: Yes

Reviewer #4: Yes

5. Review Comments to the Author

Reviewer #1: This paper seeks to evaluate the post-2024 election scenarios based on the UK party manifestos. The exercise is interesting and the authors develop a sophisticated model, but it has several major problems that make it unsuitable for publication in an academic journal. I list below my main concerns.

The first major problem is that it is not obvious what we learn from this exercise. It is interesting to derive models out of manifestos, but it is not clear why we need to know this. After reading the paper, I had a major question mark: “so what?”. Predicting how policies would look like may be interesting, but experience showed that such predictions are very often wrong and they do not match reality. We know from existing studies that a) campaign promises are not always feasible and b) circumstances change during a term in office – the dynamic of politics and society cannot be anticipated based on manifestos. In brief, it is not obvious what we gain after reading these predictions in addition to simply reading the manifestos. More important, the manifestos are programmatic promises that are rarely accomplished; in many democratic countries, political parties fulfil a small share of their manifesto promises, which means that such predictions lose their purpose entirely.

One implicit assumption of the models is that the situation during the term in office will be similar to that during the campaign in which these promises are formulated. There are many studies indicating that this is not the case, and much changes. Neither endogenous nor exogenous sources of change are included in the discussion. Moreover, even if we assume that everything stays the same in terms of the context, how do we know that the actors’ preferences do not change based on new information that they acquire or on new opportunities? The model does not account for these changes. Also, the model does not consider possibilities for coalitions or changing government partners.

Another major concern is that the manuscript does not engage with the vast literature on pledge fulfilment or strategic role of manifestos. Taking at face value the content of manifestos is not common and extensive literature shows that manifestos play a political communication role in many instances rather than aspiring to pursue specific policies.

The 2024 elections are a bad choice in terms of programmatic policies because there was a clear winner and a major loser no matter what they included in the manifestos. These are atypical elections, which were called very shorty before the date in which they were organized, parties did not have time to prepare their programmatic promises and thus there is limited credibility regarding the pursuit of some policies.

Finally, the paper does not have an analytical goal and it is mainly a description. This is useful for a course session or for journalistic accounts, but insufficient for a journal article.

Reviewer #2: This was an interesting paper and potentially a very significant contribution to the academic literature on party politics and the study of party programmes i.e. manifestos, in particular. In my view, it meets four of the journal's criteria: it presents the results of primary research, I'm not aware that the results have been published elsewhere, it meets the standards of research integrity and it adheres to community for standards for availability. It is also partially meets the remaining criteria: it is performed to a high technical standard (although there is work to be on clearly describing the approach); the conclusions are supported by the data (but the conclusions are also pointing in the wrong direction); and the article is generally clear, but it could also be much clearer.

Overall, I think this paper needs to be revised before it should be published. However, I think it should be revised for very positive reasons. I'm someone who researches manifestos, but the approach taken in this paper was extremely original. I think the authors do not do enough to capitalize on the paper's significance, and I strongly, strongly encourage them to do so. The paper should should be read and considered by all students of party politics, political campaigning, and party manifestos/platforms. Crucially, I think it has the potential to reach that audience.

[1] The paper could be very much stronger if it were reframed. It is currently framed as an almost bizarre counter-factual: what would have happened if party X, Y or Z had been elected and implemented its manifesto? This current framing is intellectually unsatisfying and further lacks realism (e.g. the way the UK system works, the importance of sequencing in the legislative timetable, the effect of events). To my mind, the obvious solution is to frame this paper about pledges and the fact that party pledges have second-order consequences on others. All studies of manifesto pledges and party ideology based on manifestos consider them narrowly in terms of their first-order effects. But this paper makes a very good case to re-think that. THIS IS VERY IMPORTANT!!! (Please excuse the emphasis...) A way to do this would be along the lines of: "Studies of manifestos pledges consider simply whether they were fulfilled or not. There is also a tendency to think about policy proposals in isolation rather than as part of joined-up programmes. This paper demonstrates the importance of joining up pledges to consider the internal consistency of manifesto commitments." I think such a framing would guarantee very high citation rates in Pol Sci journals.

[2] Related, the paper needs to do more to engage with the way political scientists have studied manifestos. Broadly speaking, there are two approaches: the pledge-fulfillment literature, and the spatial positioning of parties (i.e. the Manifesto Project). There needs to be some attempt to ground the paper in these literatures, even if the current framing is retained. Ideally, this literature would be a clearer jumping off point for the research (as per point 1).

[3] I was slightly discomfited by the decision to stick with draft proposals (Reform UK) for the use of quotes. That doesn't come across as well as it could. If the research is worth doing, it's worth doing fully. I'm also not convinced by the decision to exclude the SNP and PC, especially if the goal is to consider the second-order consequences of proposals. Remember also that the Scottish Greens are a separate party from the England and Wales Green Party. The latter do not in Scotland.

[4] Going back to point 1 and the counter-factual framing of the paper, one obvious validity test of the current approach is to consider the actual implementation of the 2019 Conservative manifesto on the environment. Did the estimate from 2019 bear closer resemblance to the actual performance of the Johnson-Truss-Sunak government? Alternatively, if these things cannot be ascertained, we're back to what the paper can get at with greater validity: the second-order impact of manifesto pledges.

[5] The paper needs to be much clearer around the methods employed. I found it difficult to follow, for example, whether the coding was computer-aided or manual, or how the scoring was arrived at. For this paper to have the reach it could potentially have, it needs to be written for those who study parties and won't have a clue about the modelling being done here. Basically, make sure the methods are written in a way that would enable, say, a historian to follow the logic.

Those are really my main thoughts. In essence: this is potentially very impactful - but it could usefully be reframed and made more accessible. I would encourage the authors to do so in for this research to fulfil the potential I think it has.

Reviewer #3: This paper aims to evaluate party manifestos in the UK 2024 general election and predict the potential policy outcomes if the manifesto promises were realized. However, several fundamental concerns make it difficult for me to fully appreciate the value of this study.

The model construction lacks transparency. The coding process appears highly subjective. The authors did not provide detailed rationales for how the edges and edge strengths were coded, nor did they explain how the quality of the coding could be evaluated. For example, do the authors assess intercoder reliability?

The authors fail to consider the underlying motivations of the parties in making their policy promises. Depending on the type of party, there are different reasons for raising certain issues in their manifestos, and these should not be treated equally.

The political context and process are completely ignored in the analysis.

Overall, it is challenging to discern what meaningful insights can be gained from this analysis.

Reviewer #4: The comments for Author are as under:

1. The introduction is informative and well-organized, but it would benefit from a clearer articulation of the study's primary research question. While the scope of the work is detailed, explicitly stating the research objectives upfront would help frame the analysis more effectively for readers.

2. While the paper references key literature on complex systems, political science, and Bayesian belief networks (BBNs), there is room for improvement in establishing a more robust link between previous research and the current study. Consider expanding the literature review to include more recent studies and critical perspectives that contextualize the use of BBNs in political scenarios.

4. The methodology is thorough, but more clarity is needed regarding the model’s assumptions. Specifically, providing a more detailed justification for the selection of node strengths and the interpretation of feedback loops within the Bayesian network would enhance the reader’s understanding of the modeling process. Additionally, the rationale for excluding smaller, devolved political parties should be more clearly addressed.

5. The results section presents valuable insights but could benefit from a more in-depth discussion on the potential limitations of the predictions. For example, consider addressing the risks associated with using simplified models for complex political scenarios. Discuss how uncertainties and assumptions in the model might affect the real-world applicability of your findings

6. Overall, the manuscript is well-written, but there are occasional grammatical errors and awkward phrasing. A thorough proofreading is recommended. The paper maintains an appropriate academic tone but occasionally lapses into informal language. Ensure that all sections adhere to formal academic writing standards

7. pplication of Bayesian belief networks to evaluate political manifestos and future policy outcomes. However, to enhance its contribution, the authors could discuss how this approach could be applied in other countries or in forecasting long-term political outcomes beyond the immediate post-election period. This would add greater depth to the potential broader impact of the study.

6. PLOS authors have the option to publish the peer review history of their article (what does this mean? ). If published, this will include your full peer review and any attached files.

**Do you want your identity to be public for this peer review?** For information about this choice, including consent withdrawal, please see our Privacy Policy .

Reviewer #1: No

Reviewer #2: No

Reviewer #3: No

Reviewer #4: **Yes: ** Dr Fouzia Amin, Assistant Professor, National Defence University Islamabad

---

## [Author Response · Author response to Decision Letter 0]

5 Jan 2025

Response to Reviewers

We would like to thank all the reviewers for their time and effort in reviewing the paper. Overall, it is clear that the purpose of the study has not come across clearly, and we have tried to address this in a newly written introduction section. Hopefully the aims and purpose of the study are much clearer (we especially thank reviewer 2 for their suggestions, and have incorporated these into the new aims). It was also clear the methods were not as clear as they could be. We have rewritten this section, but we also refer to a recently published ‘methods’ paper (by a subsection of the authors of this paper) on the BBN package used, which also provides more details. We have also reorganised the combined results and discussion into separate sections, to allow the study’s aims to be more fully discussed, and not just a party by party outcome of the results. We have also removed the second ‘environmental model’ from the analysis, as we felt this added little to the scope of the paper, and didn’t fit clearly with the new aims.

We have addressed comments one by one – but due to substantial rewriting of the manuscript, it is hard to provide quotations or line numbers of exactly what has been changed, and the rewriting has also meant the ‘track change’ manuscript is difficult to follow.

A full, point by point, response to the four reviews has been uploaded as a separate file

---

## [Decision Letter · Decision Letter 1]

31 Mar 2025

PONE-D-24-31129R1Evaluating post 2024 election scenarios for the UK based on political party manifestosPLOS ONE

Dear Dr. Stafford,

Thank you for submitting your manuscript to PLOS ONE. After careful consideration, we feel that it has merit but does not fully meet PLOS ONE’s publication criteria as it currently stands. Therefore, we invite you to submit a revised version of the manuscript that addresses the points raised during the review process.

The reviewers suggest minor revisions to further strengthen the paper. In particular, the methodology section requires a more comprehensive and detailed explanation to enhance clarity, rigor, and reproducibility of the research methods employed. Addressing these points will significantly improve the overall quality and robustness of the manuscript.

We look forward to receiving your revised manuscript.

Kind regards,

Ching-Hsing Wang, Ph.D.

Academic Editor

PLOS ONE

Journal Requirements:

Reviewers' comments:

Reviewer's Responses to Questions

**Comments to the Author**

1. If the authors have adequately addressed your comments raised in a previous round of review and you feel that this manuscript is now acceptable for publication, you may indicate that here to bypass the “Comments to the Author” section, enter your conflict of interest statement in the “Confidential to Editor” section, and submit your "Accept" recommendation.

Reviewer #3: (No Response)

Reviewer #5: All comments have been addressed

Reviewer #6: All comments have been addressed

Reviewer #7: (No Response)

2. Is the manuscript technically sound, and do the data support the conclusions?

Reviewer #3: No

Reviewer #5: Yes

Reviewer #6: Partly

Reviewer #7: Partly

3. Has the statistical analysis been performed appropriately and rigorously? 

Reviewer #3: No

Reviewer #5: Yes

Reviewer #6: Yes

Reviewer #7: I Don't Know

4. Have the authors made all data underlying the findings in their manuscript fully available?

Reviewer #3: (No Response)

Reviewer #5: Yes

Reviewer #6: Yes

Reviewer #7: Yes

5. Is the manuscript presented in an intelligible fashion and written in standard English?

Reviewer #3: Yes

Reviewer #5: Yes

Reviewer #6: Yes

Reviewer #7: Yes

6. Review Comments to the Author

Reviewer #3: The authors' response to the reviewers' comments was not comprehensive, which is understandable given the fundamental nature of some of the issues raised. While the model presents an interesting intellectual exercise, its development still appears pretty subjective in the revised version. This makes it difficult to convincingly demonstrate the scientific contribution of the paper.

Moreover, the authors treat the manifestos of all parties equally, failing to recognize the significant differences between potential ruling parties and fringe parties. Potential ruling parties may view manifestos as tools for signaling policy intentions to voters and potential coalition partners, while fringe parties may primarily use them to advance special interests or raise awareness of their positions. These differing motivations, along with the political context surrounding an election and the anticipated interactions among political actors, likely influence how parties strategically craft their manifestos. Furthermore, manifestos can be seen as initial proposals for a bargaining process, with final policies reflecting negotiations and compromises among various actors. Given these complexities, the model's assumption that a party can be the sole ruling party raises concerns about the generalizability and interpretability of the results. It remains unclear what insights can be gleaned from the analysis without considering the strategic dynamics of coalition formation and policy bargaining.

Reviewer #5: This paper “creates a complex system model of the economic, societal and environmental landscape of the UK and assesses how it would be affected if political parties enacted their 2024 general election manifesto policies in full.”

This is an interesting and unique paper (I have not reviewed a paper like this before). This is largely a thought piece that explores important counterfactuals – what might happen if part X enacted its manifesto. Though it cannot ultimately test these counterfactuals with real world empirical evidence, I think this paper is worth publishing as long as there is an openness about the study’s limitations. And I think the authors are sufficiently open about its limitations, though I would recommend a little more of such openness towards the beginning of the paper.

This paper is proposing scenarios that are very unlikely (e.g., minor parties taking full control of the UK). Nonetheless, the paper is well-written and provides many useful insights. It uses innovative methods and encourages social scientists to think about the complex connections among concepts (especially second order effects).

So let me re-emphasize that the key is to be upfront about the paper's limitations, particularly at the beginning of the paper.

Reviewer #6: As I am reviewing a revised version of the manuscript along with the earlier reviews, I see that authors put effort in making substantial changes in the manuscript which seems to lead an improved version. These changes also address some concerns I might have had. However, the lack of code and the opportunity to check it for reproducibility purposes makes it hard to arrive a conclusion even though majority of the work done with a huge undertaking.

Reviewer #7: Review of Manuscript: Evaluating Post-2024 Election Scenarios for the UK Based on Political Party Manifestos

The manuscript explores the application of a complex systems model to assess the holistic impact of UK political party manifestos for the 2024 general election. The main claim is that this modeling approach reveals both direct and indirect consequences of manifesto policies that are often obscured or misrepresented in political rhetoric. Notably, it finds that parties advocating strong tax cuts may inadvertently contribute to lower average incomes and higher inequality — a significant and counterintuitive insight.

The study is conceptually relevant and timely, particularly in the context of increasing demand for more transparent, evidence-based policy analysis. It offers a novel approach with potential utility for voters, policymakers, and political analysts.

The rationale for the study is well-articulated, and the abstract clearly identifies the gap the authors aim to address — namely, the disconnect between stated manifesto goals and actual systemic impacts. While I am not an expert in this specific modeling approach, the framing of manifestos as complex, interdependent political texts is sound and reflects current thinking in political communication and electoral studies.

That said, it would be useful for the authors to clarify how their model builds upon or diverges from previous approaches that attempt to quantify or simulate manifesto promises (e.g., spatial models, sentiment analysis, or policy simulation tools).

The manuscript is generally well-written, but I believe its readability for non-specialists should be improved. While the conclusions are somewhat lengthy, they are plausible and thought-provoking. However, the methodology section is difficult to follow in its current form. Specifically:

1) The steps taken to construct and run the model could be better explained, ideally with a synthetic visual representation (e.g., a flowchart or diagram summarizing key stages).

2) The data sources beyond the manifestos themselves are unclear. A summary table listing all data types, sources, and uses would enhance transparency and help readers better assess the robustness of the analysis.

3)The selection of manifesto topics included in the model seems potentially arbitrary, notwithstanding the authors’ efforts to justify it. Why didn’t they consider topic modelling, at least for validation purposes?

Addressing these points would go a long way toward reinforcing the credibility of the conclusions.

I believe the manuscript has potential and is generally suitable for publication in PLOS ONE, pending minor-to-moderate revisions aimed at improving clarity and transparency. As I am reviewing this in the second round and was not involved in earlier stages, I cannot speak to prior revisions, but the current version would certainly benefit from clearer presentation of methods and data.

7. PLOS authors have the option to publish the peer review history of their article (what does this mean? ). If published, this will include your full peer review and any attached files.

**Do you want your identity to be public for this peer review?** For information about this choice, including consent withdrawal, please see our Privacy Policy .

Reviewer #3: No

Reviewer #5: No

Reviewer #6: No

Reviewer #7: No

---

## [Author Response · Author response to Decision Letter 1]

28 Apr 2025

Please see attached 'Response to Reviewers' file

---

## [Editor Report · Decision Letter 2]

5 May 2025

Evaluating post 2024 election scenarios for the UK based on political party manifestos

PONE-D-24-31129R2

Dear Dr. Stafford,

We’re pleased to inform you that your manuscript has been judged scientifically suitable for publication and will be formally accepted for publication once it meets all outstanding technical requirements.

Kind regards,

Ching-Hsing Wang, Ph.D.

Academic Editor

PLOS ONE
---

## [Editor Report · Acceptance letter]

PONE-D-24-31129R2

PLOS ONE

Dear Dr. Stafford,

I'm pleased to inform you that your manuscript has been deemed suitable for publication in PLOS ONE. Congratulations! Your manuscript is now being handed over to our production team.

Kind regards,

on behalf of

Associate professor Ching-Hsing Wang

Academic Editor

PLOS ONE